# Detection and Recognition Algorithm of Arbitrary-Oriented Oil Replenishment Target in Remote Sensing Image

**DOI:** 10.3390/s23020767

**Published:** 2023-01-09

**Authors:** Yongjie Hou, Qingwen Yang, Li Li, Gang Shi

**Affiliations:** School of Information Science and Engineering, Xinjiang University, Urumqi 830017, China

**Keywords:** deep learning, pavement refill detection, MobileNetv3, CSPNet, YOLOv5s

## Abstract

In view of the fact that the aerial images of UAVs are usually taken from a top-down perspective, there are large changes in spatial resolution and small targets to be detected, and the detection method of natural scenes is not effective in detecting under the arbitrary arrangement of remote sensing image direction, which is difficult to apply to the detection demand scenario of road technology status assessment, this paper proposes a lightweight network architecture algorithm based on MobileNetv3-YOLOv5s (MR-YOLO). First, the MobileNetv3 structure is introduced to replace part of the backbone network of YOLOv5s for feature extraction so as to reduce the network model size and computation and improve the detection speed of the target; meanwhile, the CSPNet cross-stage local network is introduced to ensure the accuracy while reducing the computation. The focal loss function is improved to improve the localization accuracy while increasing the speed of the bounding box regression. Finally, by improving the YOLOv5 target detection network from the prior frame design and the bounding box regression formula, the rotation angle method is added to make it suitable for the detection demand scenario of road technology status assessment. After a large number of algorithm comparisons and data ablation experiments, the feasibility of the algorithm was verified on the Xinjiang Altay highway dataset, and the accuracy of the MR-YOLO algorithm was as high as 91.1%, the average accuracy was as high as 92.4%, and the detection speed reached 96.8 FPS. Compared with YOLOv5s, the *p*-value and mAP values of the proposed algorithm were effectively improved. It can be seen that the proposed algorithm improves the detection accuracy and detection speed while greatly reducing the number of model parameters and computation.

## 1. Introduction

Nowadays, China’s vehicle ownership is gradually increasing, and the highway and other traffic facilities have also put forward higher requirements. At the same time, along with the long-term use of China’s highways, the impact of vehicle loads, the natural environment, and other factors lead to different degrees of quality problems; highway pavement damage caused by traffic accidents frequently occurs, seriously affecting people’s travel safety. In order to prolong the service life of the highway and ensure safety performance, it is necessary to carry out stage tracking inspection to discover its loss parts in time so that certain measures can be taken effectively to prolong the life of the structure [1,2] and to prevent the occurrence of catastrophic accidents [3,4]. At present, the detection of pavement diseases in China also requires regular safety inspections by specialized pavement management departments, which then carry out maintenance and repair. Commonly used road maintenance condition inspection methods include two major categories: manual inspection and multifunctional road vehicle inspection. Although manual inspection can offer more complete statistics of pavement disease type, distribution, and size, the detection efficiency is low, there is a need to close traffic, and the detection results are subjective judgments. This method is no longer applicable to today’s pavement maintenance inspection technology needs. The multifunctional inspection vehicle detection basically does not affect the traffic, but there are defects such as disease identification, positioning, and measurement that need manual assistance. In addition, there are still problems of low efficiency and accuracy of detection. With the rapid development of the field of unmanned aircraft, the target detection technology under the UAV has increasingly become the focus of research in the field of computer vision at home and abroad, the technology has a huge practical application value in both civilian and military fields, and the application prospects are very broad. Compared with manual detection and multifunctional road inspection vehicle detection, UAV detection has the advantages of high detection efficiency and does not affect traffic. However, when using UAVs to detect small targets, it is difficult to maintain the detection rate while ensuring the detection accuracy of the target. Therefore, a method with good real-time performance and high detection accuracy needs to be designed to solve the above problems.

The development of deep learning has experienced a long start and has continued to develop rapidly over the years, receiving more and more attention from domestic and foreign researchers and scholars, and many of them have put forward innovative ideas to explore more development potentials in the field of deep learning. In terms of engineering applications, deep learning technology is integrated into all aspects of people’s daily life, such as intelligent transportation, intelligent medical care, security services, etc. Due to the rapid growth of computing power, target detection of remote sensing images has received more and more extensive attention in the field of computer vision. Initially, the application of deep learning target detection algorithms in the field of remote sensing was based on target detection models with excellent performance in generic scenes and then transplanted to the remote sensing domain images with a certain degree of adaptive optimization. Along with the great achievement of Alexnet [5] in the ImageNet visual recognition challenge, more and more deep learning models were proposed and applied to highway disease detection and protection. Common deep learning target detection methods are divided into two major categories based on the presence or absence of candidate frame generation stages:single-stage target detection algorithms and two-stage target detection algorithms. Two-stage target detection algorithms are used to generate candidate regions through specialized modules first and then make secondary corrections to the candidate frames to finally achieve the detection map, which are represented by R-CNN [6], Fast R-CNN [7], and Faster R-CNN [8]. The two-stage algorithms have slow detection speeds and a high accuracy rate. The single-stage target detection algorithm uses the idea of target regression to generate detection results by direct feature extraction calculation of the image, which is represented by YOLO [9,10,11], and SSD [12]. The single-stage algorithm is fast, but the accuracy is low.

Deep learning-based disease detection methods allow for better target feature extraction and classification. Cha et al. [13] took the form of sliding windows to segment images into blocks and used convolutional neural networks for feature extraction and classification of road pavement diseases. L Zhang et al. [14] used local block information of images and convolutional neural networks to determine the type of road diseases in a single block of images. Ale L et al. [15] compared the detection performance of SSD and RetinaNet using a dense convolutional network, a deep residual network, and a visual geometry group network as backbone networks for the road pavement distress task. The aforementioned algorithms are highly accurate in detecting highway pavement distress and protection, but they are unable to achieve accurate localization during the detection process. To be able to locate the location of road pavement diseases, Ju et al. [16] used Fast R-CNN for road disease detection, which achieved high accuracy but poor detection speed and failed to meet the real-time requirement. Rasyid et al. [17] used Faster R-CNN for disease detection, which had problems such as low detection accuracy and low efficiency. Kanaeva et al. [18] used Mask R-CNN and U-Net based for pavement crack detection with good accuracy. Mead H et al. [19] used a generative adversarial network combined with Poisson mixing to merge the disease images, while the synthesized disease images were fed into the training set, resulting in a 5% improvement in F1-Sscore metrics. Naddaf-Sh S et al. [20] implemented the EfficientDet model based on the detection of road pavement distress. Mandal V et al. [21] compared the performance of YOLOv5, EfficientDet, and CenterNet models on the pavement distress detection task. However, it is difficult for the above methods to maintain high accuracy in pavement disease detection while achieving the application requirements of real-time detection.

Our research contributions are as follows: an arbitrarily oriented target detection method based on YOLOv5 is proposed. To better match image features, a new boundary discontinuity-free rotation detector is used to solve the angular periodicity problem by transforming the angular regression problem into a classification problem, and the angular error after classification is evaluated using the long-edge representation. At the same time, the feature network of the original network was improved. A MobileNetv3-based lightweight network structure is used to replace part of the YOLOv5s backbone structure, thereby reducing the computational effort. To find regions of attention in a specific large range of images, we use a cross-stage local area network module to replace part of the MobileNetv3 network structure. Finally, by balancing the focus loss and incorporating LRM, the detection performance is improved, and the detection method is further optimized for remotely sensed images.

## 2. Methodology

### Background of YOLOv5

Target detection based on remote sensing images has high requirements for algorithm accuracy and real-time, and the target size scale varies, while the YOLO algorithm is a current detection algorithm with good real-time detection and accuracy and adapted to multi-scale target detection, and the algorithm is now increasingly used in industrial fields. Compared to previous versions, YOLOv5 features fast detection, high recognition rates, and is lightweight. Based on its small size YOLOv5s is ideally suited for deployment into embedded devices for the detection of remotely sensed image targets. Therefore, based on the comprehensive consideration of recognition accuracy and detection speed, this paper uses YOLOv5s, which has relatively high recognition accuracy and the fastest detection speed, to study and improve and optimize its network structure to achieve the demand of real-time target detection.

The YOLOv5 model differs in the depth and width of the network, which can be divided into s, m, l, and x in increasing size, all consisting of four parts: Input, Backbone, Neck, and Head. The input uses Mosaic data augmentation to randomly scale, randomly arrange, and randomly crop the image, and finally stitch the image; the adaptive initial anchor frame calculation is to set the initial anchor frame adaptively before the neural network training so as to output the predicted frame and compare it with the real frame; the image scaling adaptively adds a minimum black edge to both ends of the image and transforms it into a fixed size before feeding it into the neural network model Backbone uses Focus processing to increase the number of features in the image without changing the information in each feature, the role of the Focus is to slice the input image to ensure that the feature information is not lost without downsampling, reducing the number of network model parameters while increasing the speed of network inference, the slice operation is shown in Figure 1. The CSP structure refers to the use of residual components to make the algorithm lighter and improve the learning capability of the model while reducing the computational effort, and the SPP pooling pyramid structure enhances the ability to extract feature information from the images by increasing the perceptual field.

Neck mainly improves the residual network (ResNet) [22] and implements a multi-scale feature fusion network with FPN [23] + PAN [24] structure. The FPN layer conveys semantic information from the top down through upsampling, while the PAN layer conveys localization information from the bottom up through downsampling so that the features extracted from the backbone network and the detection network can be aggregated to improve the network’s feature fusion capability. The main function of the Head is to identify and classify the images to be detected, and there are three detection heads. Three loss functions are used to calculate the classification, localization, and confidence losses, respectively, and to improve the accuracy of the network prediction through NMS. The network structure of YOLOv5 is shown in Figure 2.

## 3. Proposed Method

The MR-YOLO model is optimized on the basis of YOLOv5s, and the improved structure is shown in Figure 2. The overall network structure consists of three parts: backbone network, feature fusion network, and output layer. The model applies the lightweight network structure MobileNetv3, replacing its backbone network (backbone) with the improved MobileNetv3-YOLOv5 structure. MobileNetv3 greatly improves the module processing speed, increases the efficiency of the overall model, and reduces the model size. The CSPNet network structure is also introduced in the backbone network to eliminate the duplicate features and computational bottlenecks generated during the computation, further ensuring the accuracy and speed of the model. Finally, the output layer replaces the original loss function on the basis of the focal loss function to solve the loss problem caused by sample imbalance. The improved structure is shown in Figure 3. 

### 3.1. Improving the MobileNetv3-YOLOv5 Model

#### 3.1.1. MobileNet Model

The core idea of the MobileNet series network is to introduce the deep separable convolution operation, which divides the standard convolution filter into two structures: deep convolution and point convolution. Compared with the classical CNN model, it mainly replaces some of the fully-connected layers to achieve the effect of reducing the computational effort and network parameters. The standard convolution process is to multiply the convolution kernel with the corresponding bit phases of the feature map and then add them together. The standard convolutional structure is shown in Figure 4.

Assuming that the standard convolution corresponds to a constant length × width of the input and output, and the convolution process is to convert the output layer with an input of $D_{F}$×$D_{F}$×M into an output layer of $D_{F}$×$D_{F}$×N, the computation of the standard convolution kernel is:(1)$$D_{F}\times D_{F}\times D_{K}\times D_{K}\times M$$

In the equation, $D_{F}$×$D_{F}$ is the length × width of the input or output layer, $D_{K}$×$D_{K}$ is the scale of the convolutional kernel filter, and M,N represents the number of input and output channels.

The depth separable convolution idea is to achieve the fusion between the information of each channel without feature combination, using only the depth convolution to perform the convolution operation for each channel alone, and then the point convolution to achieve the function by using 1 × 1 convolution. When the size of the convolution kernel of depth convolution and point convolution is $D_{K}$×$D_{K}$ and 1 × 1, the feature map is input to the depth convolution layer, and a single output is obtained after the convolution operation, which is used as the input of point convolution, and the depth feature output is obtained after the convolution operation. The depth separable convolution structure is shown in Figure 5. The computational quantities of depth convolution and point convolution are:(2)$$D_{F}\times D_{F}\times D_{K}\times D_{K}\times M\times N$$
(3)$$D_{F}\times D_{F}\times M\times N$$

The total computation of deep separable convolution is:(4)$$D_{F}\times D_{F}\times D_{K}\times D_{K}\times M+D_{F}\times D_{F}\times M\times N$$

The ratio of the computation of the depth-separable convolution to that of the standard convolution is:(5)$$\frac{D_{F}\times D_{F}\times D_{K}\times D_{K}\times M+D_{F}\times D_{F}\times M\times N}{D_{F}\times D_{F}\times D_{K}\times D_{K}\times M\times N}=\frac{1}{N}+\frac{1}{D_{K}^{2}}$$

MobileNetv3 [25] is a lightweight network architecture that has been improved from the previous two generations by the NetAdapt algorithm and the neural architecture search MNAS algorithm. mobileNetv3 first combines the deep separable convolution of MobileNetv1 [26], the inverse residual structure of MobileNetv2 [27]. Then, by introducing the light-weighted SE (Squeeze and Excitation) attention mechanism in MnasNet, the network focuses on more favorable channel information to adjust the corresponding weights of each channel; finally, the original swish function is replaced by the h-swish activation function. Finally, the original swish function is replaced by the h-swish activation function to ensure that the computation is greatly reduced under the condition of a certain number of parameters, which effectively improves the recognition accuracy of the model. The purpose of introducing the MobileNetv3 model in this paper is to reduce the computation to reduce the size of the model and improve the detection accuracy.

#### 3.1.2. CSPNet Network

Cross Stage Partial Network (CSPNet) [28] was proposed mainly to solve the problem that the use of neural networks requires a large amount of inference computation, and people need to rely on expensive computational resources. The proposed network enables the detection model to achieve richer gradient combination information, enhances the learning ability of CNN, and reduces the computational effort. By dividing the feature map of the base layer into two parts, SHORT PART, and MAIN PART, and then merging them using a cross-stage hierarchy. By splitting the gradient streams, CSPNet makes the gradient streams propagate through different network paths. Meanwhile, the CSPNet structure will be used as CSPDarknet53 and CSPResNeXt, which are often applied in the ResNet structure and Darknet53 module to reduce the computation and improve the accuracy of experimental results. In summary, it can be seen that CSPNet can effectively reduce the memory cost in the training process, greatly reduce the computation, and improve the inference speed and accuracy. The flow chart of the CSPNet module is shown in Figure 6.

### 3.2. MobileNetv3-YOLOv5 Based Network Model

This model is a feature extraction operation by replacing the Backbone backbone network of YOLOv5 with a MobileNetv3-based backbone network. Since MobileNetv3 is a lightweight model, it can be used to improve the detection speed of targets by taking advantage of its characteristics of fewer parameters, faster speed, and lower memory consumption, which can increase the operation speed while reducing the number of parameters. MobileNetv3 combines the four features of MobileNetv1 and MobileNetv2. First, the 1 × 1 convolution is up-dimensioned, the inverse residual structure of MobileNetv2 is introduced; then the multi-channel depth-separable convolution kernel is operated, and finally, the fusion of feature maps is completed after the point convolution operation to reduce the size of the network model. The experiments show that the training speed of the model is further accelerated by using the features of MobileNetv3 and the fast real-time data processing of YOLOv5 to achieve the effect of real-time. The Spatial Pyramid Pooling Network (SPPNet) is introduced to convert the feature maps obtained from deeply separable convolution operations into feature vectors matching the fully connected layers to enhance the perceptual field and thus improve the accuracy of the feature map information. The speed of the candidate frame is improved by avoiding the repeated computation of convolutional features. Finally, the Cross Stage Partial Network (CSPNet) is introduced to eliminate the duplicate features and computational bottlenecks generated during the computation, which reduces the size of the model and ensures the training speed and accuracy of the model.

### 3.3. MobileNetv3-YOLOv5 Based Network Model

The loss function used in YOLOv5s is the GIoU function [29], which is an improved version of the intersection-over-union (IoU) function [30]. IoU evaluates the performance of the target detector and is used to calculate the ratio of intersection and concurrence between the predicted and true frames. GIoU, on the other hand, introduces a penalty term on top of IoU to more The GIoU is a penalty term based on IoU to reflect the intersection ratio between the predicted frame and the real frame more accurately. The specific formulas for both are as follows:(6)$$IoU=\frac{\left|A\cap B\right|}{\left|A\cup B\right|}$$
(7)$$GIoU=IoU-\frac{\left|C-(A\cup B)\right|}{\left|C\right|}$$

In the formula, A denotes the predicted detection box, B denotes the true detection box, and C denotes the area of the smallest rectangular box that contains both the detection box and the true box, |C−(A∪B)| indicating the penalty term.

GIoU can better distinguish the position relationship between the predicted detection frame and the real detection frame when the two frames are in the case of complete intersection, GIoU = 1; when the two frames do not intersect, the farther the distance, the closer the GIoU is to −1. However, when the detection frame and the real frame appear in the special case contained in Figure 7, the penalty in GIoU is 0, which will degenerate to IoU, and cannot reflect the relative positions of the two frames. At the same time, it appears in the calculation process that the prediction box is difficult to optimize in horizontal and vertical directions, and the convergence speed is slow.

To solve the above problem, the DIoU [31] loss function is proposed, which not only can reflect the position distance under the complete inclusion of the box, but also converges faster than the GIoU function, and its expression is as follows:(8)$$DIoU=IoU-\frac{p^{2}(b,b^{gt})}{c^{2}}$$

In the formula, b, $b^{gt}$ are the positions of the centre points of the predicted and true detection frames, respectively; p denotes the Euclidean distance between the two points; c denotes the diagonal distance of the minimum closure region containing the two boxes.

CIoU [32] is an evolved version of a series of IoUs. The original DIoU loss value is relatively homogeneous and lacks a basis for judgment, so there is CIoU, which adds the penalty of aspect ratio and is no longer homogeneous but contains multiple comparison criteria such as distance to the center point and overlap area, which can ensure faster convergence of the prediction frame and higher accuracy of regression localization during training. In addition, for multi-parameter regression loss and inconsistency between objective function and evaluation index, CIoU performs best. Therefore, CIoU is selected as the frame regression loss function of the baseline algorithm in this paper. The formula of CIoU is shown in the following equation:(9)$$L_{CIoU}=1-IoU(A,B)+p^{2}(A_{ctr},B_{ctr})/c^{2}+\alpha \nu$$
(10)$$\alpha =\frac{\nu }{1-IoU+\nu }$$
(11)$$\nu =\frac{4}{\pi^{2}}(\arctan\frac{w^{gt}}{h^{gt}}-\arctan\frac{w}{h}{)}^{2}$$

In the formula, $A_{\mathrm{ctr}}$ indicates the point coordinates of the centre of the prediction frame,$B_{ctr}$ indicates the point coordinates of the centre of the real frame, $w^{gt}$ and $h^{gt}$ indicate the width and height of the real frame, and *w* and *h* indicate the width and height of the prediction frame.

In the algorithm of this paper, the way in which the rotation frame is defined is determined. The baseline algorithm structure needs to be optimized to support the prediction of rotating frames, and a new rotation angle prediction channel *θ* is added to the original Head structure to achieve the prediction of rotating frames. As shown in Figure 8, the improved Head structure is illustrated in detail. The remote sensing image is extracted by the feature extraction network layer and Neck layer to obtain the final detection layer P, which has a channel dimension of 3 × (C + 6). The CIoU Loss is also used to optimize the position and shape of the long-edge definition method [33] to obtain the final prediction results.

Among them, the sample matching and border encoding process during training is exactly the same as the baseline YOLOv5. The main difference is that θ is also encoded. This paper makes use of the characteristics of the output value interval of the Sigmoid function to further prevent the network structure angle prediction value from exceeding the defined range of the long-edge representation. The optimized loss component consists of four parts: confidence loss, class classification loss, edge regression loss, and theta angle classification loss. For real scenarios where there are samples that are difficult to be learned, Focal Loss [34] has a good role in regulating the samples. By using a balancing factor α as well as a hyperparameter adjustment factor γ, the final Focal Loss formula output is shown below. The probability corresponding to the predicted class of the model is represented by *p,* and y is the sample class label, which takes the value of 1 or 0.
(12)$$FL(p,y)=\left\{\begin{array}{c} -\alpha {\left(1-p\right)}^{\gamma }\log_{a}p,y=1\\ -(1-\alpha )p^{\gamma }\log_{a}(1-p),y=0 \end{array}\right.$$

The main training process of this network is divided into extracting the category data and regression data first, coding the regression variables in order to reduce the difficulty of network learning, and finally calculating the loss. For YOLOv5, which is a multi-category label detection algorithm, the rate NMS algorithm is used to remove redundant frames to achieve the final result in order to improve the detection rate of blurred categories in aerial scenes. Finally, for the effect of foreground-background samples in the actual scene, LRM [35] can filter out low-loss values and leave high-loss values that benefit the detector. In this paper, a feature map ranking factor is used to find the most suitable high-loss value. For example, if b = 0.1 and the feature map size is 3 × 40 × 40, the 480 cells with the highest detection loss are selected, and the remaining cells are excluded from the backpropagation process, further gaining improved detection performance. The final loss was calculated as follows.
(13)$$Loss_{\mathrm{total}}=\lambda Loss_{cls}+\beta Loss_{CIoU}+\alpha Loss_{angle}+\gamma Loss_{theta}$$

## 4. Experiments

This paper is implemented based on the Pytorch 1.7 deep learning framework, using GPU for training. The model uses Adam as the optimizer during training, and the specific configuration of the experimental environment is shown in Table 1 based on the existing configuration.

The relevant experimental parameters are as follows: lr denotes the learning rate, momentum denotes the learning rate momentum, weight_decay denotes the weight decay coefficient, epoch denotes the training batch, and batchsize denotes the batch size, as shown in Table 2.

### 4.1. Experimental Dataset

There is currently no large-volume traffic road patching dataset available at home or abroad, and this experiment uses a homemade dataset from the Big Data Smart IoT Lab at Xinjiang University. The dataset was captured on a road section in Altay, Xinjiang, and the whole process was carried out by a UAV. To ensure the accuracy of the detection results, 2500 images containing road patch marks were captured, each with a size in the range of 4000 × 4000 to 5472 × 5472. The 2500 images were randomly divided into a training set and a validation set in the ratio of 8:2, with 2000 images for the training set and 500 images for the validation set. The images were resized to 640 × 640 before training, and all image labels were chosen to be labeled with directed data and converted to long-edge representation for training and prediction. Figure 9 below shows a section of the image containing traces of oil filler.

### 4.2. Long Edge Marking Method

Due to the existence of complex scenes and diversity in target scales in the oil replenishment dataset, a large number of small, cluttered, and rotating targets are very sensitive to angular changes, while it is difficult to accurately detect the area information of the target object using horizontal bounding boxes, which cannot meet the detection requirements, so the rotating box annotation method is adopted in this paper. According to the previous literature on rotating frame detection, there are two main types of common methods for defining arbitrary rotating frames, namely the five-parameter definition method [36] and the eight-parameter quadrilateral definition method [37,38,39]. The five-parameter definition method can be described as the representation of a rotating frame with angular information. Opencv and long-edge representations are common five-parameter representations. Opencv contains five main parameters [*x*,*y*,*w*,*h*,*θ*]. Among them, *θ* indicates the direction of the range of 360 degrees, 180 degrees, and 90 degrees three because the research object of this paper does not need to strictly determine the object’s positive and negative, so the main take is 180 degrees and 90 degrees two. The following Figure 9a indicates that the angled label varies within 90 degrees, the angle is at an acute angle to the *x*-axis, the angle label information is independent of the length and width, the detection method has interchangeability between the long and wide sides, i.e., w is either long or wide. Although the angular training loss is reduced by compressing the angular range, it also increases the difficulty of regression of the aspect and further reduces the convergence speed of the neural network due to the periodicity problem of angular rotation, which is not satisfactory in practical application scenarios. Figure 10b indicates that the angle labels vary within 180 degrees, which defines the long side as the angle between the x-axes so that the aspect is always unique. This approach has a large loss of neural network angle during training, but the actual angle difference is small. Angle regression in this approach is more difficult due to the periodicity of the angle.

Despite the competitive performance detection advantages achieved by parametric regression-based rotation detection methods for vision, these methods have largely suffered from boundary problems. Based on the above analysis, some methods have been proposed to address these problems, such as IoU-Smooth L1 Loss [40] to eliminate the problem of sudden increases in boundary loss by introducing an IoU constant factor to modularise the rotational loss [41] to increase the boundary constraint. However, these methods are still regression-based detection methods and do not provide a solution to the root cause. To address a series of problems with sudden increases in loss values and difficulty in learning the target angle by the network, the angle regression approach is converted to a categorical form by introducing CSL labels while using a Gaussian function as a window function so that the loss value between the actual angle of the target and the Gaussian label of the model’s predicted angle is calculated when calculating the angle loss. By converting the angle regression problem to a classification task, the total category is 180, which is a multi-category task, so BCEWithLogitsLoss is used as the loss function for the angle. In this paper, the window radius r is 6. The specific expression for the ring smoothing label is as follows:(14)$$CSL(x)=\left\{\begin{array}{l} ae^{-{\left(x-b\right)}^{2}/2c^{2}},\theta -6<x<\theta +6\\ 0,otherwise \end{array}\right.$$

In the formula, *a*, *b* and *c* are real constants, and *a* > 0. *θ* is the angular value of the rotation.

### 4.3. Experimental Evaluation INDEX

Remote sensing image target detection usually uses several metrics to evaluate algorithms, and this experiment uses mean Average Precision (mAP), Precision (P), and Frame Per Second (FPS) as the main evaluation criteria. Precision represents the proportion of correctly branched oiling targets. For example, if there are ten oiled targets in an image and the algorithm detects only five targets, three of which are oiled targets, then the precision is 60%. The recall represents the proportion of the number of positive samples in the network that are detected. For example, if there are ten oil-filled targets in an image, and the algorithm detects five oil-filled targets, then the recall is 50%. Mean accuracy (mAP) indicates the accuracy of multiple sets of data and is averaged. Frames per second (FPS) indicates how many images can be processed per second. Their calculation process is shown in Formulas (15)–(18).
(15)$$P=\frac{TP}{TP+FP}$$
(16)$$R=\frac{TP}{TP+FN}$$
(17)$$AP=\int_{0}^{1}PdR$$
(18)$$mAP=\frac{1}{N}\sum_{C=1}^{C}AP_{C}$$

In the formula, *FP* indicates the number of false detections in the test; *TP* indicates the number of correctly detected samples in the test; FN indicates the number of correct samples predicted as false samples in the test; $AP_{C}$ represents the *AP* value of the Cth category.

### 4.4. Experimental Results and Analysis

#### 4.4.1. Effectiveness Experiments

In order to fully validate the effectiveness of the module improvements in this paper, experimental analysis was conducted on the road refueling dataset of the Big Data Smart IoT Lab at Xinjiang University to verify the importance of each of the proposed components. The experiments use YOLOv5s as the baseline model and embed each component into the baseline model in turn, where improvement a indicates that the Backbone backbone network of YOLOv5 is replaced with the MobileNetv3-based backbone network to perform the feature extraction operation; improvement b indicates that the CSPNet network structure is introduced in the backbone network to eliminating duplicate features and computational bottlenecks generated during computation; improvement c indicates the use of long-edge representation to achieve rotation of the target frame, replacing the effect of periodic variations induced by the regression problem on network training with classification ideas. The same hyperparameters and training techniques were used in each set of experiments, and the results of the ablation experiments are shown in Table 3.

From Table 3, it can be seen that the improved method proposed in this paper has different degrees of improvement in algorithm accuracy and average precision compared to the original YOLOv5s algorithm on the homemade dataset, where the CSL module represents a long-edge representation to achieve rotation of the target frame, using classification ideas instead of the impact of periodic changes induced by the regression problem on the network training, the module detects the best improvement, with algorithm accuracy and average accuracy improved by 4.6%, 4%, and 4.3%, respectively. The addition of the MobileNetv3 module reduces accuracy and average precision. However, when combined with the CSPNet module, the accuracy and average precision improved by 2.6%, 3.7%, and 5.9%. The effectiveness of the different improvement methods proposed in this paper was again demonstrated through ablation experiments, where the accuracy and average accuracy of the MR-YOLO algorithm proposed in this paper were higher than the detection results of the base value YOLOv5s, where the *p*-value was improved by 8.4% and the mAP by 8.3% and 7.4%, respectively. It is verified that the algorithm proposed in this paper can better deal with the detection and identification of road refills based on remote sensing images.

#### 4.4.2. Performance Comparison

In addition, in order to further verify the effectiveness of the algorithm proposed in this paper, the algorithm proposed in this paper was also compared with other remote sensing image target detection algorithms for experimental analysis and validated on the validation set. The results are shown in Table 4. According to the comparative experiments in Table 4, it can be seen that the algorithm proposed in this paper achieves the best results in terms of *p*-value and mAP-value.

From Table 4, it can be concluded that the algorithm in this paper outperforms several other detection algorithms in general when performing detection on the oil replenishment dataset. Compared to the YOLOv3 model, the introduction of MobileNetv3 and the combination of CSPNet, etc., will greatly improve the detection accuracy but is not the optimal solution, and the average mean and FPS are not the most suitable to meet the needs of high accuracy detection and real-time detection. The method proposed in this paper combines the above-mentioned shortcomings, and through comparison experiments, it can be seen that the MR-YOLO detection algorithm has great advantages in the detection and identification of pavement oil fillings. The accuracy of this algorithm is 91.4%, the average accuracy is 92.4%, and the average detection time is 96.8 ms per frame on the oil fill data set.

In this paper, the refueling dataset is selected for testing under different conditions, which include normal altitude in the same flight environment and flight altitude in different environments. The influencing factors include different scales, different target sizes, and different background conditions to compare and validate. The YOLOv5s algorithm was used for comparative analysis with the algorithm in this paper, and the test results are shown in Figure 11a–d.

Some of the detection results of the proposed algorithm in this paper are shown in Figure 10 above. It can be seen that the detection results of the algorithm proposed in this paper are better than the YOLOv5s algorithm in a normal environment. Figure 10a,b shows the proposed algorithm in this paper has high detection accuracy, can detect more ultra-small targets, and has higher advantages in the detection scenario for small targets. As for the detection of the oil replenishment category with arbitrary directional rows and varying target scales, the effect of the prediction frame of the algorithm proposed in this paper is closer to the real shape while further solving the influence of background on target classification, and the comparison results are shown in Figure 10c,d. It is further demonstrated that the algorithm in this paper is more accurate than YOLOv5 detection and reduces the problem of expanding the target detection area by the horizontal boundary prediction frame and causing losses caused by the later road assessment and maintenance stages; this detection performance is further enhanced in the detection of remote sensing images with obvious scale variations.

By comparing the detection results of the algorithm in different environments at different flight heights, as shown in Figure 11, it can be concluded that the detection effect of the algorithm proposed in this paper is better and can be more suitable for the detection needs of the technical state of the road. From Figure 11a,b, it can be concluded that in the scenario where the target image is not horizontal, the use of the rotation detection method proposed in this paper can improve the detection accuracy and better match the real target frame, which further provides accurate data for the subsequent real road inspection and maintenance. At the same time, it can be seen from Figure 12c,d that in the detection of ultra-small targets, the algorithm proposed in this paper has a better detection effect and can clearly detect the target in the ultra-long range environment.

## 5. Conclusions

For UAV-based pavement refill detection and recognition, the common algorithms for natural scenes cannot meet the problems of detection accuracy and detection rate, as well as the demand of practical application scenarios; this paper proposes an improved MR-YOLO detection algorithm for YOLOv5s. The algorithm introduces the MobileNetv3 lightweight network structure and enhances the extraction capability of feature maps by incorporating this module in the feature extraction network, reducing the model size and increasing the detection rate while using the CSPNet network structure to eliminate computational bottlenecks and making the algorithm better meet the demand for real-time detection. In addition, a rotating frame is used to cope with the difficulty of obtaining orientation information about the target object’s motion in a horizontal bounding frame, while a circular labeling method is used to reduce the impact of variation in loss values for the periodicity problem caused by angular regression. Finally, the focal loss function is improved to further enhance the detection accuracy. The effectiveness of the MR-YOLO network model is verified by scar identification experiments under the road refill dataset. The comparison of experimental results with other algorithms shows that the proposed algorithm in this paper has improved the *p*-value, mAP-value, and FPS are all higher than those of the YOLOv3 model. However, the angles of the oil-filled images captured in this dataset are relatively single, and the trained model has a high false detection rate for oil-filled images from different angles, while the FPS is not the best at present. In order to further enhance the generalization ability of the model so that it can meet the detection of scars from different angles and improve the detection speed, we will try to apply the model to the training of road oiling dataset with different angles and more complex backgrounds, so that it can be optimized in terms of detection rate to meet the detection requirements.

## Figures and Tables

**Figure 1 sensors-23-00767-f001:**
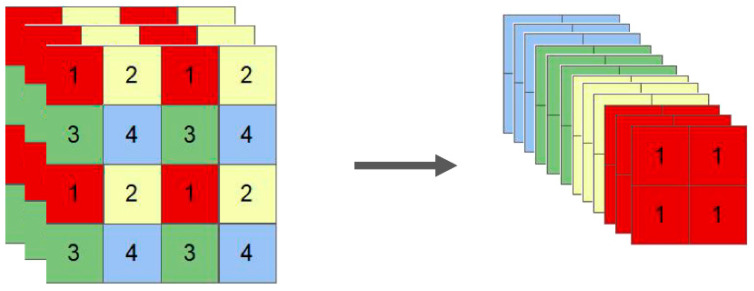
Focus slicing process.

**Figure 2 sensors-23-00767-f002:**
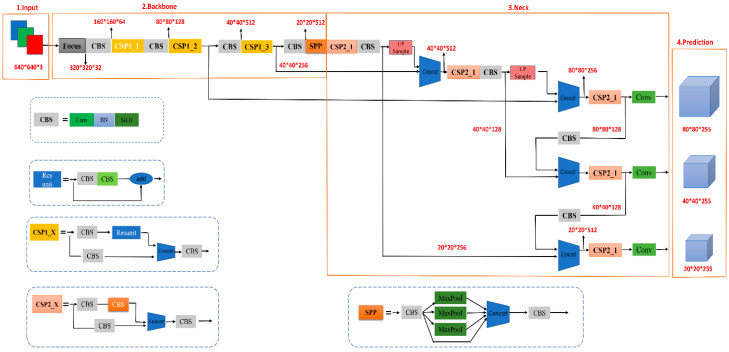
YOLOv5s network structure. It is mainly composed of four parts: input, backbone network, neck network, and prediction network.

**Figure 3 sensors-23-00767-f003:**
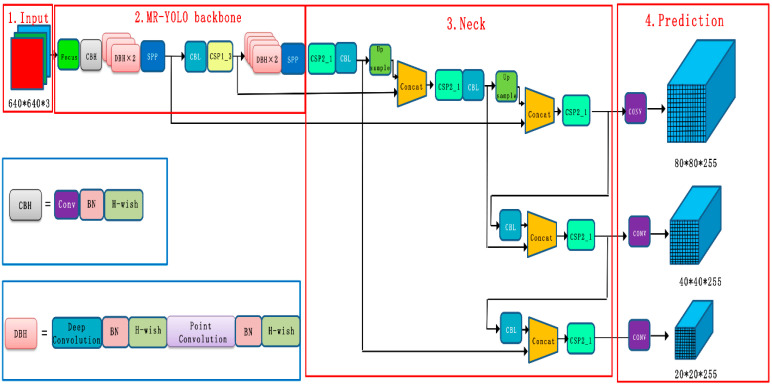
Network structure diagram of improved YOLOv5. The components DBH + CBH form the feature extraction network part. CBH: consists of three components Conv + BN + H-wish activation function. DBH: consists of two components depth separable convolution + point convolution.

**Figure 4 sensors-23-00767-f004:**
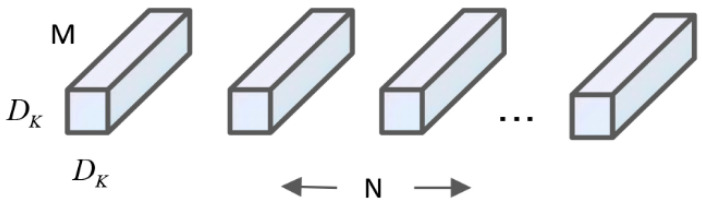
Standard convolution structures.

**Figure 5 sensors-23-00767-f005:**
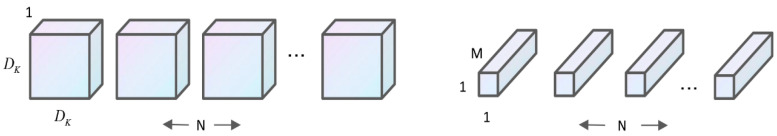
Depth separable convolution.

**Figure 6 sensors-23-00767-f006:**
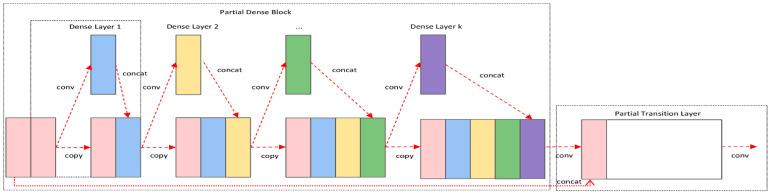
CSPNet module flow chart.

**Figure 7 sensors-23-00767-f007:**
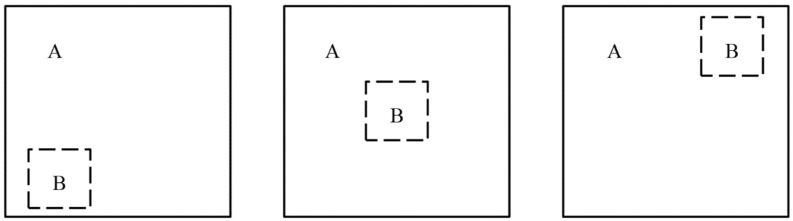
Schematic diagram of the relationship between the predicted frame and the real frame position.

**Figure 8 sensors-23-00767-f008:**
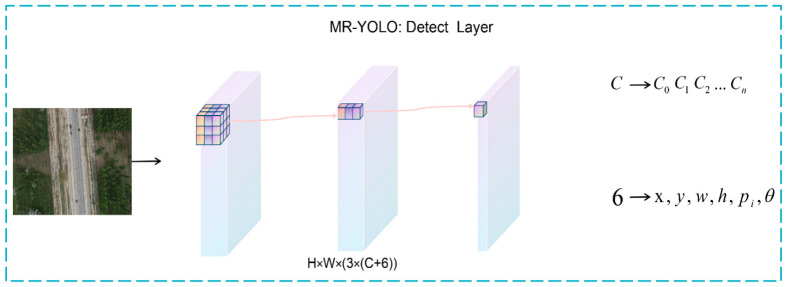
MR-YOLO: Head Detail Inspection Chart.

**Figure 9 sensors-23-00767-f009:**
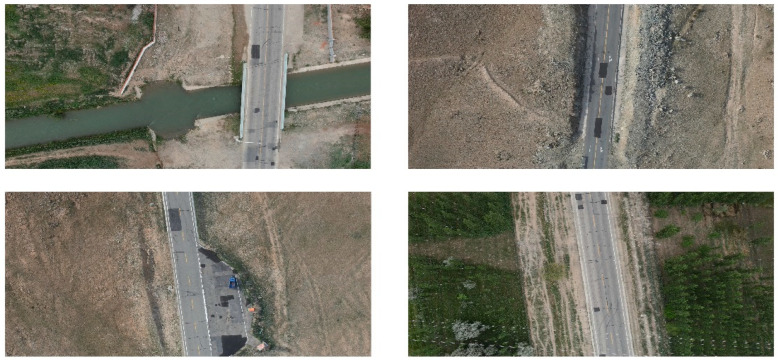
Sample refill dataset.

**Figure 10 sensors-23-00767-f010:**
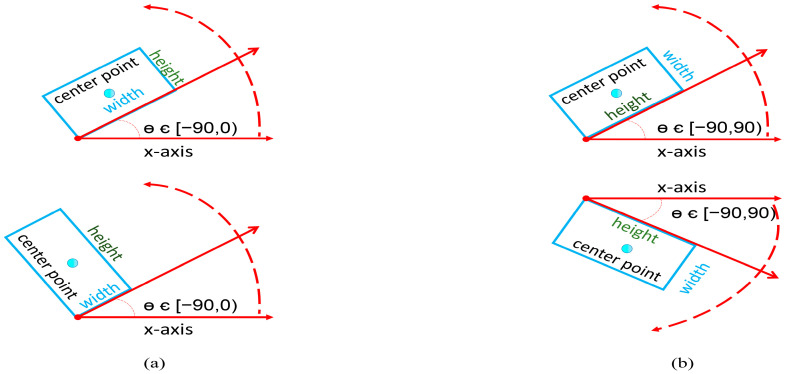
Method of defining the rotating frame for different angle ranges. (**a**) Five-parameter method with an angular range of 90°. (**b**) Five-parameter method with an angular range of 180°.

**Figure 11 sensors-23-00767-f011:**
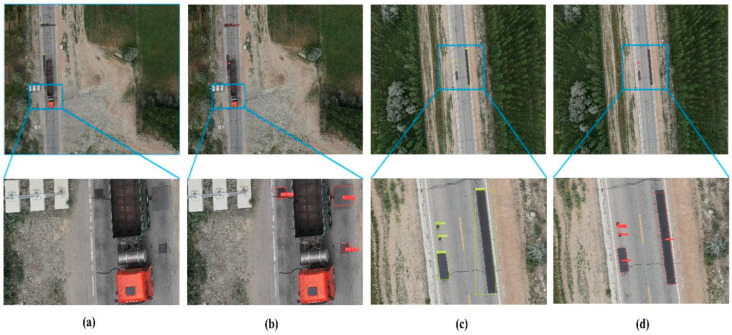
Comparison graph of algorithms for the same conditions of flight altitude. (**a**,**c**) MobileNetv3 + YOLOv5s + CSL test results. (**b**,**d**) MR-YOLO test results.

**Figure 12 sensors-23-00767-f012:**
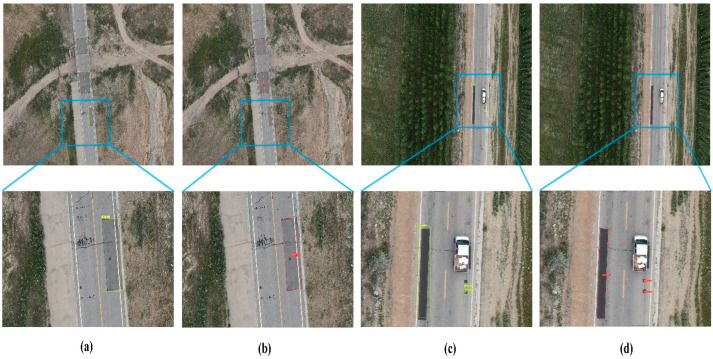
Comparison graph of algorithms for the different conditions of flight altitude. (**a**,**c**) MobileNetv3 + YOLOv5s + CSL test results. (**b**,**d**) MR-YOLO test results.

**Table 1 sensors-23-00767-t001:** Experimental environment configuration.

Attribute	Value
OS	Ubuntu 20.04
GPU	NVIDIA RTX 3090Ti
Memory	24 GB
Deep learning framework	Pytorch1.7
CUDA	11.0

**Table 2 sensors-23-00767-t002:** Network parameters of MR-YOLO.

Lr0	Momentum	Weight_Decay	Epoch	Batchsize
0.01	0.937	0.0005	300	32

**Table 3 sensors-23-00767-t003:** Experimental comparisons of each combination in the feature extraction network.

Method	MobileNetv3	CSPNet	CSL	P/%	mAP@0.5/%	mAP@0.95/%
YOLOv5				82.7	84.1	64.5
a	√			81.9	83.5	58.3
b		√		84.5	86.2	67.1
c			√	87.3	88.1	68.8
d	√	√		85.3	87.8	70.4
e	√		√	89.4	89.6	71.3
Ours	√	√	√	91.1	92.4	71.9

**Table 4 sensors-23-00767-t004:** Performance comparison of different algorithms.

Model	P	mAP@0.5/%	mAP@0.95/%	FPS/f*s−1
YOLOv3	83.2	85.3	65.1	75.9
MobileNetv3 + YOLOv3	85.1	86.2	57.7	78.3
MobileNetv3 + YOLOv3 + CSPNet	86.5	87.4	66.2	74.7
YOLOv3-Tiny	85.3	87.2	65.3	80.4
YOLOv5s	82.7	84.1	64.5	131.7
MobileNetv3 + YOLOv5s	83.4	85.5	66.3	128.5
MobileNetv3 + YOLOv5s + CSPNet	86.8	88.3	67.5	112.1
MobileNetv3 + YOLOv5s + CSL	87.1	89.3	70.9	105.8
SwinTransformer + YOLOv5s	85.4	86.7	58.3	59.5
SwinTransformer + YOLOv5s + CSPNet + CSL	90.1	91.3	70.9	65.8
Ours	91.1	92.4	71.9	96.8

## Data Availability

Not applicable.

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
