# Peer review of "Detection and Recognition Algorithm of Arbitrary-Oriented Oil Replenishment Target in Remote Sensing Image"

_sensors, 2023, doi:10.3390/s23020767_

Round 1
Reviewer 1 Report
This paper proposes a new target detection algorithm to improve the detection effect of existing remote sensing images. This study is interesting and valuable, but needs to be further revised before publication
(1) The author has proposed a detection algorithm based on YOLO framework, but there is less discussion about YOLO in the introduction. In order to highlight the contribution proposed in this paper, more discussion about YOLO is necessary. (e.g., https://doi.org/10.1111/mice.12500 and https://doi.org/10.3390/app11020813)
(2) What are the advantages of MobileNetv3? Why the method proposed in this paper adopts it.
(3) What criteria are used to determine the selection of network parameters?
(4) The author explained the computer configuration. I was a little confused. The NVIDIA GTX 2080Ti GPU handles 1024×1024 images. Can its FPS reach 96.8?
Reviewer 2 Report
The MR-YOLO algorithm was proposed in this manuscript to improve the accuracy and speed of pavement oil mark detection. MobileNetv3 lightweight network structure was introduced to reduce the model volume and improve the detection rate. SPPNet network structure was instructed to remove redundant feature options and improve the detection accuracy of the model. It can be seen from the experimental results of MR-YOLO and other algorithms that the P-value and mAP value of the algorithm proposed in this paper is improved compared with YOLOv5s. Although the FPS is higher than the YOLOv3 model, it is not the best at present. Now, the faster YOLOv7 is coming, and you can improve your model by considering YOLOv7.
In addition, here are some small tips you need to concern below:
1) In Line 293, The colon is followed by an extra period.
2) Table 1, Line 359 and Line 360 are overlapped with the table.
3) Figure 1 does not appear to be your original, so please make the right citation.
4) Your title "Oil Replenishment" does not seem appropriate.
Round 2
Reviewer 1 Report
No comment